# Cycloartenyl Ferulate Is the Predominant Compound in Brown Rice Conferring Cytoprotective Potential against Oxidative Stress-Induced Cytotoxicity

**DOI:** 10.3390/ijms24010822

**Published:** 2023-01-03

**Authors:** Hongyan Wu, Toshiyuki Nakamura, Yingnan Guo, Riho Matsumoto, Shintaro Munemasa, Yoshiyuki Murata, Yoshimasa Nakamura

**Affiliations:** 1School of Food Science and Technology, Dalian Polytechnic University, Dalian 116034, China; 2Graduate School of Environmental and Life Science, Okayama University, Okayama 700-8530, Japan

**Keywords:** cycloartenyl ferulate, antioxidative effect, cytoprotective potential, heme oxygenase-1, nuclear factor erythroid 2-related factor 2

## Abstract

Since brown rice extract is a rich source of biologically active compounds, the present study is aimed to quantify the major compounds in brown rice and to compare their cytoprotective potential against oxidative stress. The content of the main hydrophobic compounds in brown rice followed the order of cycloartenyl ferulate (CAF) (89.00 ± 8.07 nmol/g) >> α-tocopherol (αT) (19.73 ± 2.28 nmol/g) > γ-tocotrienol (γT3) (18.24 ± 1.41 nmol/g) > α-tocotrienol (αT3) (16.02 ± 1.29 nmol/g) > γ-tocopherol (γT) (3.81 ± 0.40 nmol/g). However, the percent contribution of CAF to the radical scavenging activity of one gram of whole brown rice was similar to those of αT, αT3, and γT3 because of its weaker antioxidant activity. The CAF pretreatment displayed a significant cytoprotective effect on the hydrogen peroxide-induced cytotoxicity from 10 µM, which is lower than the minimal concentrations of αT and γT required for a significant protection. CAF also enhanced the nuclear factor erythroid 2-related factor 2 (Nrf2) nuclear translocation coincided with the enhancement of the heme oxygenase-1 (*HO-1*) mRNA level. An HO-1 inhibitor, tin protoporphyrin IX (SnPP), significantly impaired the cytoprotection of CAF. The cytoprotective potential of CAF is attributable to its cycloartenyl moiety besides the ferulyl moiety. These results suggested that CAF is the predominant cytoprotector in brown rice against hydrogen peroxide-induced cytotoxicity.

## 1. Introduction

Rice (*Oryza sativa* L.) is one of the most commonly consumed grains in the Asian regions, accounting for approximately 90% of the world’s consumption [1]. Brown rice is the most famous of the rice varieties for having a wide range of physiological properties benefiting human health. Several reports have shown that the brown rice extract exerted health-promoting effects, such as anti-obesity [2], cholesterol-lowering [3] and anti-inflammatory [4] effects. Our previous study found that the ethanol extract of brown rice and its representative lipophilic antioxidant, α-tocopherol (αT), could alleviate the hydrogen peroxide-induced loss of cell viability, probably associated with the enhancements of the different phase 2 drug-metabolizing enzyme expressions [5]. However, the contribution of αT to the antioxidant capacity of one gram of the whole brown rice was approximately 3%, which was quite restricted. In addition to the insufficient αT content in brown rice, the different patterns of the enhanced gene expression between the brown rice and αT treatments led us to the speculation that other phytochemicals might be attributed to a cytoprotective potential of brown rice. Therefore, to provide biological evidence suggesting the desirable utility of the rice functions, studies to quantify the other predominant compounds in the whole rice and to compare and estimate their contributions to the antioxidative capacity of brown rice are imperative.

Although the content of bioactive substances in rice differs with rice varieties and extraction conditions, not only vitamin E derivatives (tocopherols and tocotrienols) and ferulic acid (FA), but also γ-oryzanols (ORYs), including cycloartenyl ferulate (CAF), are known as the quantitatively predominant compounds in brown rice, exhibiting significant antioxidant activities [6,7]. Previous studies showed that brown rice contained approximately 262.0–627.0 μg/g of total ORYs [8], 161.4–374.8 μg/g of total FA, and 13.7–36.6 μg/g of total vitamin E derivatives [7]. A recent study reported that αT and γ-tocopherol (γT) displayed a similar efficacy to attenuate the generation of mitochondrial reactive oxygen species (ROS), whereas γT exhibited a more inhibitory effect on the expression of pro-apoptosis proteins than αT [9]. ORYs, composed of the esters of FA with several triterpene alcohols or phytosterols, have been expected as hybrid phytochemicals with diverse pharmacological properties, including an antioxidant effect [10] and lipid-lowering potential [11]. Among the identified ORY derivatives, CAF, 24-methylenecycloartanyl ferulate, campesteryl ferulate, and β-sitosteryl ferulate composed more than 80% portion of the total ORYs [6]. It has been found that CAF was one of the most commonly detected ORYs in rice, exhibiting both antioxidant and antimicrobial capacities [12]. As for its antioxidant potential, the co-oral administration of FA and ORY enhanced the antioxidative enzyme activities in the mouse ethanol-elicited liver injury model [13]. However, the contribution of these predominant bioactive compounds to the cytoprotective potential in brown rice remains to be evaluated.

Considerable studies supported the idea that when an organism is exposed to exogenous factors, such as alcohol intake, radiation, and environmental toxins, the excess of ROS produced can disturb the redox homeostatic balance and cause detrimental modifications to biomolecules that are associated with the development of chronic and degenerative diseases [14]. The nuclear factor erythroid 2-related factor 2 (Nrf2) serves as one of the pivotal transcriptional factors, participating in attenuating the oxidative stress resulting from ethanol [15], ionizing radiation [16], and heat stress [17]. Nrf2 migrates into the nucleus upon exposure to oxidative stress and is bound to antioxidant response elements (AREs) in the promoter regions of phase II drug metabolism genes. These genes encoded cytoprotective proteins with a variety of beneficial functions. For instance, heme oxygenase-1 (HO-1) catalyzes the degradation of heme to produce a ferrous ion and small direct antioxidants, carbon oxide (CO) and biliverdin/bilirubin. Quinone oxidoreductase 1 (NAD(P)H dehydrogenase [quinone] 1; NQO1) catalyzes the detoxification of the electrophilic quinones into their corresponding hydroquinones, reducing the possibility of sulfhydryl deprivation associated with the generation of reactive species [18].

To the best of our knowledge, no studies have focused on comparing the contributions of the major bioactive compounds to the antioxidant capacity of whole brown rice. We thus estimated the contributions of the major bioactive compounds in the brown rice using the in vitro radical scavenging assay and reverse-phase high-performance liquid chromatography (HPLC) analysis. The comparative study of the cytoprotective potentials of these compounds then identified CAF as the predominant compound in brown rice regarding resistance to the hydrogen peroxide-induced oxidative damage. The molecular mechanism underlying its antioxidative cytoprotection was further classified. In addition, the contribution of different components to the cytoprotective capacity of CAF was also evaluated.

## 2. Results

### 2.1. Estimation of the Contribution of Major Phytochemicals to Antioxidant Capacity in Brown Rice

ORYs and FA, as well as vitamin E derivatives, are considered the major contributors to the antioxidant effect of brown rice, which are closely associated with the disease-preventing effects of rice consumption [19]. Therefore, we first determined the content of these representative lipophilic compounds in the whole brown rice, which is one standard variety (Hinohikari) cultivated in Japan, by a HPLC analysis (Appendix A). As shown in Table 1, the amount of CAF in the whole grain of brown rice (89.00 ± 8.07 nmol/g) was approximately five-fold higher than that of αT (19.73 ± 2.28 nmol/g). The average amounts of γ-tocotrienol (γT3) (18.24 ± 1.41 nmol/g) were quite similar to that of α-tocotrienol (αT3) (16.02 ± 1.29 nmol/g) and much higher than that of γT (3.81 ± 0.40 nmol/g). However, the content of FA in the 100% ethanol extract of brown rice was too low to be detected. We next estimated the contributions of these predominant compounds to the total radical scavenging activity of one gram of brown rice by comparing the concentrations with their Trolox equivalent (TE) values. Interestingly, the percent contribution of the most abundant compound, CAF (3.1%), to the radical scavenging activity of one gram of whole brown rice was quite similar to that of αT (3.0%), αT3 (2.9%), and γT3 (3.3%), whereas γT accounted for only around 0.6% of the total activity (Table 1). The estimated TE value for CAF or vitamin E derivatives in the whole brown rice are shown in Appendix A. These estimations suggested that, although CAF is the richest antioxidative component in brown rice, its contribution to the antioxidant capability of brown rice in vitro is comparable to that of each compound belonging to the vitamin E family. This might be due to the fact that the free radical-scavenging activity of CAF is quite lower than those of the vitamin E derivatives (Appendix A).

To further examine which is the most effective component in brown rice in the cultured cell model, we then compared the antioxidant effects of these major phytochemicals against the hydrogen peroxide-induced cytotoxicity in Hepa1c1c7 cells. As shown in Figure 1, the 24-h treatment of each compound dose-dependently recovered the decrease in cell viability induced by oxidative stress and their significant protective effects were observed at each of the different concentrations. The significant protection of γT (Figure 1b) was observed from 25 μM, quite similar to that of αT [5]. On the other hand, the remarkable effects of αT3 and γT3 were seen at the lower concentration, 5 μM (Figure 1c,d), implying that the unsaturated phytyl side chain might enhance the cytoprotective potential of tocotrienols. CAF also showed effective cytoprotection from a relatively lower concentration, 10 μM (Figure 1a). Combined with the quantitative analysis results, we speculated that CAF might be the predominant component responsible for the total cytoprotective effect of brown rice.

### 2.2. Cycloartenyl Ferulate (CAF) Is a Potential Cytoprotector against Hydrogen Peroxide

Since CAF is the predominant compound in brown rice to have a cytoprotective potential against the hydrogen peroxide-induced oxidative damage in cells, we next evaluated the cytoprotective mechanisms. As shown in Figure 2b,c, the treatment of Hepa1c1c7 cells with CAF for 6 h significantly increased the mRNA level of *HO-1*, whereas no significant enhancement was observed for the expression of *NQO1*. We next examined whether the CAF treatment for 6 h suppressed the hydrogen peroxide-induced cytotoxicity. As shown in Figure 2d, the CAF treatment for 6 h significantly protected the cells from 50 μM to 100 μM. Since the Nrf2 nuclear translocation plays an important role in the induction of various cytoprotective proteins, including HO-1 [20], we checked the modulating effect of CAF on the nuclear Nrf2 level. As shown in Figure 3, the treatment of CAF for 6 h increased the nuclear Nrf2 level in a dose-dependent manner, suggesting that CAF activated the Nrf2 nuclear translocation and thus triggered the gene expression of downstream cytoprotective proteins (Figure 2b). These results suggested that the Nrf2-dependent enhancement of the *HO-1* expression might, at least partly, contribute to the cytoprotective potential of CAF.

### 2.3. The Cytoprotective Effect of CAF Was Counteracted by an Heme Oxygenase-1 (HO-1) Inhibitor, Tin Protoporphyrin IX (SnPP)

To gain further evidence for the involvement of HO-1 in the cytoprotective mechanism, we employed tin protoporphyrin IX (SnPP), a representative HO-1 activity inhibitor. Although the cell viabilities decreased by hydrogen peroxide alone to 76%, CAF at 10 μM and 25 μM significantly protected the cells from hydrogen peroxide, with the cell viabilities recovering to 83% and 88%, respectively (Figure 4). On the other hand, the pretreatment of SnPP at the non-toxic concentration impaired the CAF-induced cytoprotection; the cell viability decreased to 74% and 78%, respectively. These results strongly supported the idea that the HO-1-dependent metabolic products, such as the antioxidative biliverdin/bilirubin and carbon monoxide, might be involved in the cytoprotective mechanism of CAF.

### 2.4. Cycloartenyl Moiety Mainly Contributes to the Cytoprotective Potential of CAF

Since CAF is an ester compound composed of FA and cycloartenol (CA), we evaluated the contribution of either the ferulyl or cycloartenyl moiety to the cytoprotective potential of CAF. Since CAF (50 μM, 6 h) enhanced the *HO-1* mRNA expression (Figure 2b) and significantly protected the cells from hydrogen peroxide (Figure 2d), the modulating effects of FA and CA on the *HO-1* gene expression were compared under the same treatment condition (50 μM, 6 h). Acetone, as well as ethanol, were used as a vehicle for these compounds and thus as a control. As shown in Figure 5, CA showed a significant inducibility of the *HO-1* mRNA levels comparable to CAF, but FA did not significantly enhance it at the same concentration. This finding suggested that the cycloartenyl moiety, more than the ferulyl moiety, contributes to the cytoprotective effect of CAF.

## 3. Discussion

In the current study, CAF was identified as the predominant compound in the whole brown rice in terms of its quantity as well as cytoprotective efficacy. The quantitative experiments revealed that the CAF content was approximately five times higher than that of the most abundant vitamin E derivative, αT (Table 1), consistent with a previous study showing that ORYs are the most abundant bioactive compounds in rice [21]. The CAF amount (89.00 nmol/g) in whole brown rice examined in the present study was consistent with the previous report [22], whereas Cho et al. reported the higher amount of CAF (300 nmol/g) [23]. This difference might be associated with the different rice variety and extraction methods, including solvents. Of all vitamin E derivatives, αT and γT3 were the major tocopherol and tocotrienol in the whole brown rice, respectively, which was in good agreement with the previous reports [24]. The contents of αT (19.73 nmol/g), γT (3.81 nmol/g), αT3 (16.02 nmol/g), and γT3 (18.24 nmol/g) were similar to the previous report, which determined the amounts of vitamin E derivatives in eight brown rice varieties [7]. However, the content of FA in the 100% ethanol extract of brown rice was too low to be detected. This might be because most of the FA exists in the form of the conjugated phenolics in whole brown rice [7]. The contribution of CAF to the in vitro radical scavenging activity of the whole brown rice was almost the same as those of the vitamin E derivatives, since the free radical-scavenging activity of CAF is much lower (Table 1 and Appendix A). Contrarily, the concentration of CAF required for cytoprotection against the hydrogen peroxide-induced cytotoxicity was lower than those of tocopherols and not very different from those of tocotrienols (Figure 1). The radical scavenging capability attributed to the phenolic group of CAF has been determined to contribute to its cytoprotective effect against oxidative stress [25]. However, the limited contribution of CAF to the in vitro antioxidant activity of whole brown rice (Table 1) led us to the speculation that antioxidative mechanisms other than the direct radical-scavenging effect might contribute to its cytoprotective action.

We demonstrated here that CAF significantly increased the mRNA level of *HO-1* (Figure 2b), the small molecular weight antioxidant-producing enzyme, at concentrations similar to that required for cytoprotective effects in resistance to oxidative damage (Figure 2d). Moreover, the protective effect of CAF on the hydrogen peroxide-induced oxidative damage was cancelled in the presence of the HO-1 inhibitor, SnPP (Figure 4). Thus, the antioxidative products of the HO-1 enzymatic reaction, such as biliverdin/bilirubin and CO, might participate in the cytoprotective potential of CAF. A recent study demonstrated that the internal CO produced by HO-1 is capable of attenuating the oxidative damage induced by UVB exposure, possibly via enhancing the protein expression levels of superoxide dismutase along with the regulation of the mitochondrial respiration process [26]. In the lipopolysaccharide-induced lung damage model, the oxidative injury in mouse alveolar macrophage cells was effectively impaired by the activation of the phosphatidylinositol 3-kinase (PI3K)/Akt pathway, followed by the HO-1/CO axis [27]. In addition, the bilirubin nanoparticles were one of the efficient tools for amelioration of the oxidative injury caused by hydrogen peroxide [28] and ischemia [29]. Collectively, these findings also implied that its catalytic products of HO-1 might play a crucial role in the cytoprotective mechanism of CAF.

Recently, a large group of food-derived phytochemicals have been proposed to target the Nrf2-mediated signaling pathways to exert cytoprotective potentials, such as benzyl isothiocyanate [18], allyl isothiocyanate [30], curcumin [31], and quercetin [32]. In the current study, the cytoprotective mechanism of CAF might involve the induction of HO-1 via the Nrf2 (Figure 3). In addition to the Nrf2-mediated pathway, other signaling pathways might participate in the protective effects of ORYs. For instance, the protective efficacy of ORY on the acetaminophen-induced oxidative injury might mainly be ascribed to modulating the nuclear factor kappa B (NF-κB) signaling pathway, in addition to the adenosine monophosphate-activated protein kinase (AMPK)/glycogen synthase kinase 3β (GSK3β)/Nrf2 pathway [33]. The administration of ORY elevated the antioxidative capacity of the muscles in mice, possibly through activating the peroxisome proliferator-activated receptor delta (PPARδ)- and estrogen-related receptor gamma (ERRγ)-dependent pathways, repressing transforming growth factor (TGF)-β pathway [34]. The attenuation of ethanol-induced oxidative damage was accomplished by the modulating effects of ORY supplementation on the mitogen-activated protein kinase (MAPK)-related pathways [35]. Anyway, although the interplay among the diverse signaling pathways are reported to engage in the cytoprotective mechanism of CAF, the Nrf2 mediated pathway might, at least in part, play an important role.

Previous studies have also reported that CAF possesses a wide range of health-promoting effects. For instance, CAF is capable of acting as an anti-allergic compound to reduce the occurrence of mast cell degranulation [36]. Yasuda et al. [37] found that CAF served as an antioxidant agent to repress the amounts of intracellular ROS in human colon cancer cell lines. CAF was also reported to enhance the expression of multidrug resistance protein 1 to attenuate the apoptosis in human kidney cell lines induced by paraquat [38]. Furthermore, the application of CAF-containing ORYs could effectively reduce the level of amyloid-β plaques in the mouse sporadic Alzheimer’s disease model [39], which might be beneficial to regulate cerebral hypometabolism. Our results also indicated that CAF has a potential to enhance the endogenous cytoprotection system. Therefore, CAF might be a prospective agent in health promotion.

When considering the chemical structure, we identified the cycloartenyl moiety as the major structural factor for the cytoprotective potential of CAF, but the ferulyl moiety might play a relatively minor role (Figure 5). A recent animal study regarding the absorption and metabolism of ORYs suggested that some quantities of ORYs would be metabolized into the FA and phytosterols to exert physiological effects after oral administration, in addition to their intact forms distributed into the blood and organs [40]. Another study using the human hepatoma cell lines demonstrated that FA could attenuate the hydrogen peroxide-induced oxidative damage through activating the Nrf2 signaling pathway [41]. Moreover, FA has the potential to scavenge the free radicals [42] and to ameliorate the multidrug resistance by regulating the NF–κB signaling pathway [43]. Therefore, the ferulyl moiety of CAF could not be ruled out in the mechanism underlying its cytoprotective efficacy. On the other hand, evidence also delineated the cytoprotective potential of phytosterols might be due to their structural similarity to the cholesterol, which has scavenging capabilities for free radicals [44], or due to mediating the signaling pathways to inhibit the generation of ROS [45]. CA was reported to ameliorate the oxidative damage caused by benzoyl peroxide or UVB radiation through restoring the activities of the cytoprotective proteins, such as glutathione-S-transferase and quinone reductase [46]. In addition, CA significantly inhibited the growth and colony-forming capabilities of the glioma cells, possibly through the modulation of the oxidative stress-related p38 MAPK signaling pathway [47]. These findings indicated that the cycloartenyl moiety of CAF might also have the potential to induce certain signaling pathways via targeting multiple intracellular molecules, including Nrf2.

## 4. Materials and Methods

### 4.1. Materials and Chemicals

Brown rice was provided by the Satake Corporation (Hiroshima, Japan). D-αT was obtained from TCI Chemicals Company (Tokyo, Japan). D-γT, CAF, and CA were purchased from FUJIFILM Wako Pure Chemical Corporation (Osaka, Japan). αT3, γT3, and FA were purchased from Funakoshi Co., Ltd. (Tokyo, Japan). Other chemicals were obtained from FUJIFILM Wako Pure Chemicals Corporation (Osaka, Japan), as well as Nacalai Tesque (Kyoto, Japan).

### 4.2. Preparation of the Rice Extract

The preparing process of the extract of whole brown rice, which is one standard variety (Hinohikari) cultivated in Japan, was performed as previously reported [5].

### 4.3. DPPH Assay

The determination of the DPPH radical scavenging activity was carried out according to the protocols in our previous study [5]. Briefly, 0.6 mL of the rice ethanol extract or the test bioactive compound dissolved and diluted in ethanol was mixed with an equal volume of a DPPH ethanol solution (0.1 mM), then the mixture was placed at room temperature in the dark for 30 min. The absorbance of the reaction solution was then measured at 520 nm. Trolox was utilized as a standard antioxidant. The radical scavenging activity of each compound was calculated as moles of DPPH radical scavenged by 1 mol of Trolox (Trolox equivalent antioxidant capacity (TEAC)) using a calibration curve obtained from Trolox in 100% ethanol. The radical scavenging capacity of the test compound in one gram of whole brown rice was expressed as nmol TE/g rice.

### 4.4. HPLC Determination of Major Compounds in the Brown Rice

The concentrations of each predominant bioactive compound in the brown rice samples were analyzed by HPLC as previously reported [5]. In brief, the major compounds in the brown rice were separated on an Inertsil C8 column (4.6 mm i.d. × 150 mm, GL Sciences, Tokyo, Japan). Elution process was performed by isocratic elution (water:methanol = 7:93 for αT, γT, αT3, γT3, and CAF and water:methanol = 60:40 for FA) with the flow rate of 1.0 mL/min. A fluorescence detector with excitation at 295 nm and emission at 325 nm was used to determine αT, γT, αT3, and γT3. A UV detector with detective wavelength setting at 290 nm and 320 nm was used to determine CAF and FA, respectively. We identified these compounds by their retention times and co-chromatography, then determined their contents by a comparison with authentic compounds (Appendix A). The minimal detection limit for αT, γT, αT3, and γT3 were 10 nM and that for CAF and FA was 300 nM. The TE value of each compound was calculated using a calibration curve established by measuring the antioxidant activity using authentic standards of each compound at the different concentrations. The contribution of each compound to the antioxidant capacity of whole brown rice was calculated by comparing the TE value of each compound estimated from its content in the whole brown rice with the TE value exhibited by the whole brown rice.

### 4.5. Cell Cultures

The mouse hepatoma cell line Hepa1c1c7 was purchased from the American Type Culture Collection and cultured in the minimum essential medium-α (MEMα, Gibco), supplemented with 10% fetal bovine serum, penicillin (100 U/mL), and streptomycin (100 μg/mL) at 37 °C in a 5% CO_2_ containing atmosphere.

### 4.6. MTT Assay for Cell Viability Determination

The cells were seeded at a density of 2 × 10^4^ cells per well in the 96-well plate for 24 h, following by treating with each bioactive compound or ethanol as a vehicle (final concentration, 0.1%) for the indicated time points. Before the 6-h of exposure to the one hundred micromolar of hydrogen peroxide, the treated cells were washed with phosphate-buffered saline one time. After the cells incubating with the MTT solution (0.5 mg/mL) for 2 h, the insoluble formazan crystals were dissolved in 2-propanol, and the absorbance was determined at 570 nm using a microplate reader (Benchmark Plus, Bio-Rad Laboratories, Hercules, CA, USA). The cell viability values were expressed as the percentages over the corresponding controls.

### 4.7. RNA Extraction and Reverse Transcription-PCR Analysis

The cells were precultured in a 60 mm dish at a density of 5.0 × 10^5^ for 24 h, following by exposing to the different concentrations of test compound for the indicated time points. Total RNA was extracted by the TRIzol reagent according to the manufacturer’s manual. The total RNA (5 µg) was reverse transcribed to cDNA using the ReverTra Ace (Toyobo, Osaka, Japan). Then, the PCR amplification was accomplished using Taq polymerase and specific primers for *HO-1* and *NQO1*. The primer sequences and the expected PCR product sizes are as follows: *mHO-1*, (F) 5′-ACATCGACAGCCCCACCAAGTTCAA-3′ and (R) 5′-CTGACGAAGTGACGCCATCTGTGAG-3′ (22 cycles, product size 203 bp); *mNQO1*, (F) 5′-TCGAAGAACTTTCAGTATCC-3′ and (R) 5′-TGAAGAGAGTACATGGAGCC-3′ (23 cycles, product size 290 bp); *mβ-actin*, (F) 5′-GTCACCCACACTGTGCCCATCTA-3′ and (R) 5′-GCAATGCCAGGGTACATGGTGGT-3′ (16 cycles, product size 455 bp). The amplified PCR products were separated on an agarose gel (2%), stained with ethidium bromide, and visualized with a LAS3000 image analyzer (FujiFilm, Tokyo, Japan). The relative densities of bands were measured using the Image J program.

### 4.8. Western Blotting Analysis

Hepa1c1c7 cells (2.0 × 10^5^) were cultured with CAF for 6 h. The separation of nuclear fraction was performed, as previously reported [20]. The relative densities of the bands were measured using the Image J program.

### 4.9. Statistical Analysis

All values were expressed as the mean of at least three independent experiments ± standard deviation. The statistical significance was determined through Student’s paired two-tailed t-test or one-way ANOVA followed by Tukey’s HSD by SPSS version 16.0 (IBM, Chicago, IL, USA). A *p*-value < 0.05 was considered significant in all comparisons.

## 5. Conclusions

In conclusion, CAF was identified as a major phytochemical in brown rice in terms of its ability to increase the resistance against hydrogen peroxide. The HO-1-dependent mechanism through the Nrf2 signaling pathway might, at least in part, be involved in the CAF-induced cytoprotection against the hydrogen peroxide. As for the structural aspect, the cycloartenyl moiety might play the more important role in the cytoprotective potential of CAF than ferulyl moiety. Because CAF is abundant in the whole brown rice, this study may provide a biological basis for the development of CAF-based nutraceuticals in the food industry. On the other hand, the cultured mouse hepatocyte model has some limitations, i.e., not reflecting the characteristics of human ones, and not being the chronic stress model. Additionally, the Hepa1c1c7 cells do not reflect the characteristics of intact hepatocytes because this cell line originated from hepatocellular carcinoma. In addition, the concentrations of CAF required for its cytoprotection (10 μM) might be supraphysiological. Therefore, future studies will be concerned with the significance of CAF in in vivo rodent models, as well as human hepatocyte models. It is also necessary to further understand the detailed intracellular pathways involved in the rice phytochemical-induced cytoprotection and the synergistic effects of typical rice compounds.

## Figures and Tables

**Figure 1 ijms-24-00822-f001:**
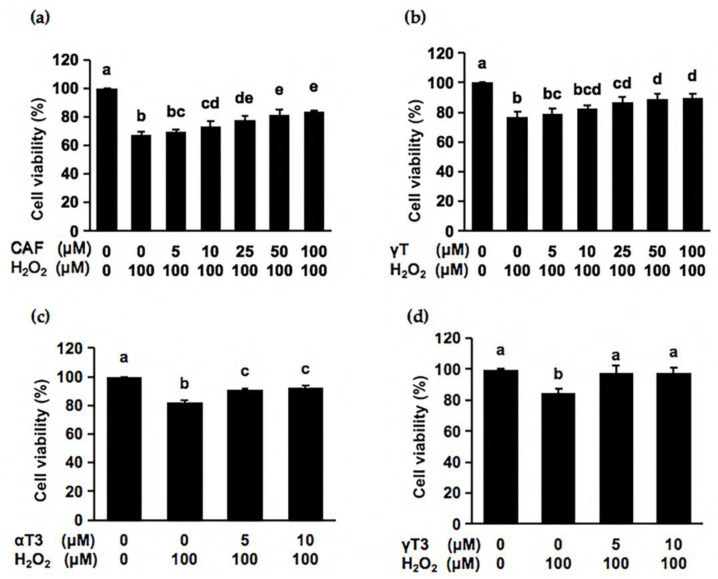
Protective effect of the main phytochemicals of brown rice in Hepa1c1c7 cells. Hepa1c1c7 cells were preincubated with cycloartenyl ferulate (CAF) (**a**), γ-tocopherol (γT) (**b**), α-tocotrienol (αT3) (**c**) or γ-tocotrienol (γT3) (**d**) at the indicated concentrations for 24 h, then treated with hydrogen peroxide (100 μM) for 6 h. Cytotoxicity was evaluated using a MTT assay. Values were means ± SD of three independent experiments. The different letters above the bars indicated significant differences among the treatments for each condition (*p* < 0.05).

**Figure 2 ijms-24-00822-f002:**
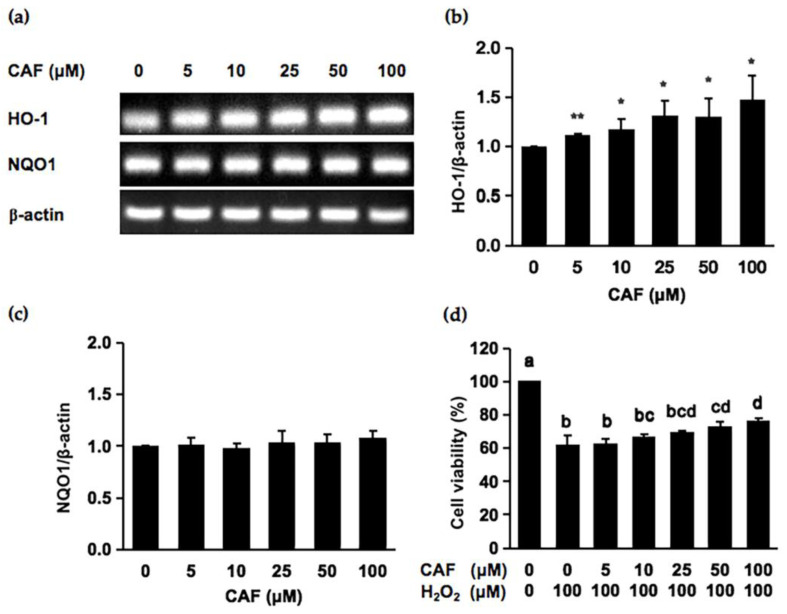
Modulating effect of CAF on the gene expression of cytoprotective proteins and the hydrogen peroxide-induced cytotoxicity in Hepa1c1c7 cells. (**a**–**c**) Hepa1c1c7 cells were pretreated with various concentrations of CAF for 6 h and the mRNA levels of heme oxygenase-1 (*HO-1*) and quinone oxidoreductase 1 (*NQO1*) were analyzed by RT-PCR. All values were expressed as means ± SD of three separate experiments. ** *p* < 0.01; * *p* < 0.05 vs. control. (**d**) Hepa1c1c7 cells were pretreated with various concentrations of CAF for 6 h, followed by treatment with 100 μM of hydrogen peroxide for 6 h. Cell viability was measured using a MTT assay. All values were expressed as means ± SD of three separate experiments. The different letters above the bars indicated significant differences among the treatments for each condition (*p* < 0.05).

**Figure 3 ijms-24-00822-f003:**
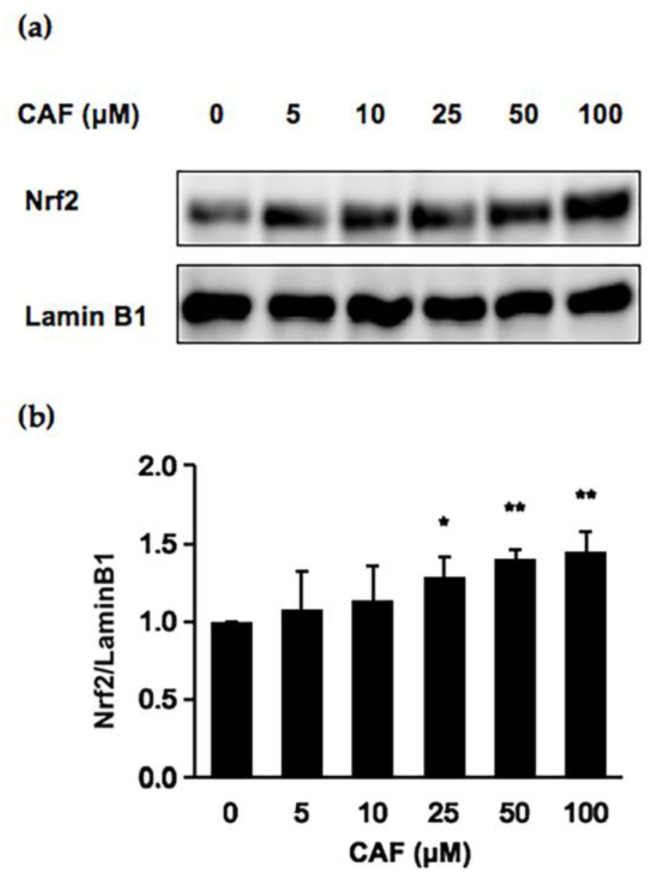
Modulating effects of CAF on the nuclear factor erythroid 2-related factor 2 (Nrf2) level (**a**,**b**). The nuclear fraction was separated from Hepa1c1c7 cells treated with CAF for 6 h, then a Western blot analysis for each protein was carried out. All values were expressed as means ± SD and analyzed by Student’s t-test (* *p* < 0.05, ** *p* < 0.01 compared to control).

**Figure 4 ijms-24-00822-f004:**
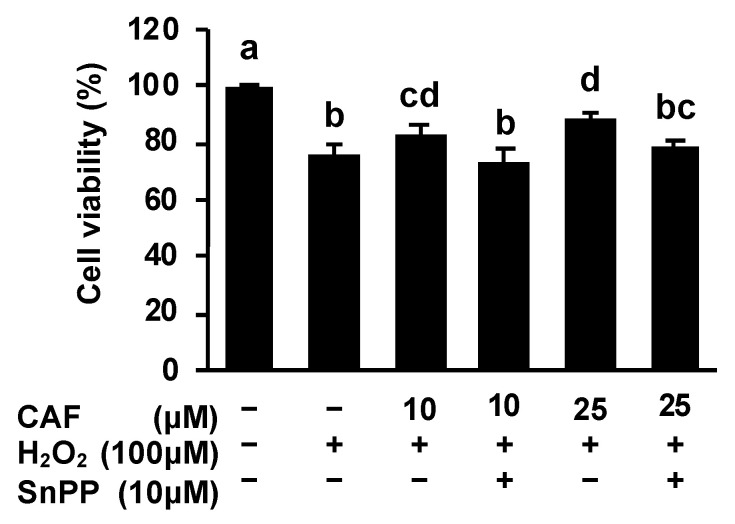
Tin protoporphyrin IX (SnPP), the representative HO-1 inhibitor, impaired the CAF-induced cytoprotection against hydrogen peroxide in Hepa1c1c7 cells. Cells were pretreated with SnPP for 1 h, followed by a 24-h incubation with CAF, then hydrogen peroxide exposure for 6 h. Cell viability was measured using a MTT assay. Values were means ± SD of three independent experiments. The different letters above the bars indicated significant differences among the treatments for each condition (*p* < 0.05).

**Figure 5 ijms-24-00822-f005:**
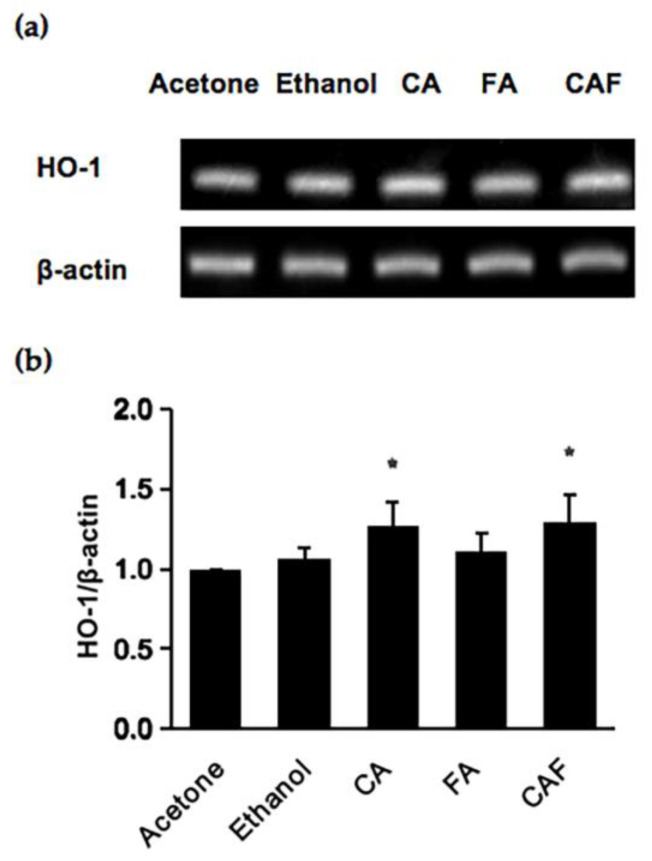
Comparison of the effects of cycloartenol (CA), ferulic acid (FA) and CAF on the HO-1 gene expression (**a**,**b**). Hepa1c1c7 cells were treated with 50 μM of CAF, CA, or FA for 6 h, and the mRNA level of HO-1 was analyzed by RT-PCR. Acetone was the vehicle for CAF and ethanol for CA and FA. All values were expressed as means ± SD of five separate experiments. * *p* < 0.05 vs. control.

**Table 1 ijms-24-00822-t001:** The concentrations of the major phytochemicals in the brown rice and their percent contribution to the antioxidant capacity of whole brown rice ^a^.

Rice Sample	αT (nmol/g) (Percent Contribution to Antioxidant Capacity of Rice)	γT (nmol/g) (Percent Contribution to Antioxidant Capacity of Rice)	αT3 (nmol/g) (Percent Contribution to Antioxidant Capacity of Rice)	γT3 (nmol/g) (Percent Contribution to Antioxidant Capacity of Rice)	FA (nmol/g) (Percent Contribution to Antioxidant Capacity of Rice)	CAF (nmol/g) (Percent Contribution to Antioxidant Capacity of Rice)
BR	19.73 ± 2.28 (3.01 ± 0.22%)	3.81 ± 0.40 (0.59 ± 0.03%)	16.02 ± 1.29 (2.91 ± 0.22%)	18.24 ± 1.41 (3.33 ± 0.26%)	N.D ^b^	89.00 ± 8.07 (3.09 ± 0.16%)

The percent contribution of each chemical to the antioxidant capacity of whole brown rice was calculated by comparing the Trolox equivalent (TE) value of each compound estimated from its content in the whole brown rice with the TE value exhibited by the whole brown rice. ^a^ Results are expressed as mean ± SD. (n = 4). ^b^ Not detected.

## Data Availability

The data presented in this study are available from the corresponding author upon reasonable request.

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
