# Peer review of "Cycloartenyl Ferulate Is the Predominant Compound in Brown Rice Conferring Cytoprotective Potential against Oxidative Stress-Induced Cytotoxicity"

_ijms, 2023, doi:10.3390/ijms24010822_

Round 1

Reviewer 1 Report

Abstract

Lines 17-18: Amounts should be included.

Introduction

Line 31: Replace “Oryza sativa L.” by Oryza sativa L.”

Line 49 – 53: Authors said that γ-oryzanols, ferulic acid, tocopherols, and tocotrienols are quantitatively predominant in brown rice. However, any amount reported by previous studies was presented. Considering that some readers could not be experts on brown rice's phytochemical composition, I suggest that it be briefly presented, emphasizing γ-oryzanols, ferulic acid, tocopherols, tocotrienols, and other bioactive compounds previously identified in brown rice with antioxidant activity.

Line 59 – 60: What were the other ORY identified in brown rice? Amounts?

Material and Methods

Line 297 – 300: Was a calibration curve performed?

Line 311 – 313: How was the TE of each compound evaluated? Were standards of each compound used to determine it? How was the calculation of the contribution to the antioxidant capacity of whole brown rice performed?

Line 320: Was the density 2 x 104 or 2 X 105?

Results

Table 1: Authors should explain (in the footnotes) how the percent contribution to the antioxidant capacity of whole brown rice was calculated.

Figure 2 d: Data shown in Figure 1a and Figure 4 indicated that the pretreatment with 10 uM of CAF had a statistically significant effect on suppressing the hydrogen peroxide-induced cytotoxicity. However, data depicted in figure 2d indicated that this significant effect only occurs at concentrations equal to or higher than 50 uM. How do you explain these differences? Could it be related to different pretreatments times?

Discussion  

Data shown in table 1 should be discussed. Were they in agreement with data from previous studies?

Conclusion

The conclusion depicts the present study's limitations very well and suggests good opportunities for future research.

Author Response

  1. Lines 17-18: Amounts should be included.

As suggested, we have added amounts of major compounds in brown rice in the Abstract section (Lines 17-19).

Introduction

  1. Line 31: Replace “Oryza sativa L.” by “Oryza sativa L.”

Thank you very much for pointing out the error.  We have replaced “Oryza sativa L.” by “Oryza sativa L.” in the Introduction section (Line 32).

  1. Line 49 – 53: Authors said that γ-oryzanols, ferulic acid, tocopherols, and tocotrienols are quantitatively predominant in brown rice. However, any amount reported by previous studies was presented. Considering that some readers could not be experts on brown rice's phytochemical composition, I suggest that it be briefly presented, emphasizing γ-oryzanols, ferulic acid, tocopherols, tocotrienols, and other bioactive compounds previously identified in brown rice with antioxidant activity.

According to this suggestion, we have emphasized the predominant bioactive compounds identified in brown rice with antioxidant activity and briefly presented the total contents of γ-oryzanols, ferulic acid, and vitamin E derivatives in brown rice in the Introduction section (Lines 52-55) with citation of the corresponding references (Lines 437-440).

  1. Line 59 – 60: What were the other ORY identified in brown rice? Amounts?

In response to this comment, we have added the major ORYs identified in brown rice and their proportions to the total ORYs (Lines 61-63), citing the corresponding reference (Line 435-436).

Material and Methods

  1. Line 297 – 300: Was a calibration curve performed?

According to the reviewer’s suggestion, we have added the description of calibration curve in the Materials and Methods section (Lines 323-324).

  1. Line 311 – 313: How was the TE of each compound evaluated? Were standards of each compound used to determine it? How was the calculation of the contribution to the antioxidant capacity of whole brown rice performed?

In response to these comments, we have rewritten the descriptions about how to calculate the contribution of each compound to the antioxidant capacity of whole brown rice in the Materials and Methods section (Lines 337-342).

  1. Line 320: Was the density 2 x 104 or 2 X 105?

In the MTT assay for cell viability determination, the cell density we have used was 2 × 104 cells per well in the 96-well plate.

  1. Table 1: Authors should explain (in the footnotes) how the percent contribution to the antioxidant capacity of whole brown rice was calculated.

We agree with the reviewer’s idea that we should explain in the footnotes how the percent contribution to the antioxidant capacity of whole brown rice was calculated.  In response to this point, we have added the explanation in the footnotes of Table 1 (Lines 120-122).

  1. Figure 2 d: Data shown in Figure 1a and Figure 4 indicated that the pretreatment with 10 uM of CAF had a statistically significant effect on suppressing the hydrogen peroxide-induced cytotoxicity. However, data depicted in figure 2d indicated that this significant effect only occurs at concentrations equal to or higher than 50 uM. How do you explain these differences? Could it be related to different pretreatments times?

In Figure 1a and Figure 4, after Hepa1c1c7 cells were pretreated with CAF at indicated concentrations (0-100 microM) for 24 h and washed out, the cells were challenged by hydrogen peroxide. The significant cytoprotection under this situation was observed from 10 microM. However, in Figure 2d, when the co-incubation time of CAF with cells was reduced to 6 h before hydrogen peroxide exposure, the significant cytoprotective effect of CAF was observed at relatively higher concentrations. We considered that effective concentration of CAF required for the cytoprotection might be related to its co-incubation time with Hepa1c1c7 cells. For the short-term pretreatment with CAF, higher concentration (at least 50 microM) is needed for the cytoprotective potential of this compound.

Discussion 

  1. Data shown in table 1 should be discussed. Were they in agreement with data from previous studies?

According to this suggestion, we have added discussion about the data in Table 1 (Lines 214-225), citing the corresponding references (Lines 437-438, 478-484).

Conclusion

The conclusion depicts the present study's limitations very well and suggests good opportunities for future research.

              Thank you very much for the reviewer for his/her careful reading of the manuscript and constructive suggestions. We believe that the paper is significantly improved and hope that the corrections made together with the attached reply satisfy the concerns raised.

Reviewer 2 Report

Please take into account the the following points:

* The originality of the manuscript should be clearly indicated in the end of the introduction section.

* The identification of the compounds separated through HPLC analysis should be clearly specified in the M & M section (subheading 4.4).

* HPLC chromatographic profile showing the evidence of the compounds explored in the manuscript should be added in the results section.

Author Response

  1. The originality of the manuscript should be clearly indicated in the end of the introduction section.

As suggested, we have added the descriptions about the originality of the manuscript in the Introduction Section (Lines 84-85).

  1. The identification of the compounds separated through HPLC analysis should be clearly specified in the M & M section (subheading 4.4).

As suggested, we have added the details of compound identification by HPLC analysis in the Materials and Methods section (Lines 334-336).

  1. HPLC chromatographic profile showing the evidence of the compounds explored in the manuscript should be added in the results section.

According to the suggestion of reviewer, we have added HPLC chromatographic profile in Figure S1 (Line 100).

Reviewer 3 Report

Dear authors,

Your work adds to the mechanistic links between a dietary staple and eventual health outcomes.

Given that the paper does mention the general health benefits of rice, you might consider adding a single paragraph / overview of CAF pharmacology, ie mast cell stabilisation, apoptosis induction, amyloid chemistry etc

I also think you should acknowledge the Yasuda et al '19 paper, https://www.jstage.jst.go.jp/article/jos/68/8/68_ess19054/_article

You may have a specific and valid reason not to do so?

Author Response

Your work adds to the mechanistic links between a dietary staple and eventual health outcomes. Given that the paper does mention the general health benefits of rice, you might consider adding a single paragraph / overview of CAF pharmacology, ie mast cell stabilisation, apoptosis induction, amyloid chemistry etc

I also think you should acknowledge the Yasuda et al '19 paper, https://www.jstage.jst.go.jp/article/jos/68/8/68_ess19054/_article

You may have a specific and valid reason not to do so?

According to the suggestion of this reviewer, we have added one more paragraph focusing on the pharmacology of CAF (Lines 271-281) with citation of the related references (Lines 514-523).

Round 2

Reviewer 1 Report

Authors performed all changes suggested.